

**FATE OF DISSOLVED ORGANIC MATTER ACROSS THE PERMAFROST-NEARSHORE WATER**
**CONTINUUM : ROLE OF THE INTERTIDAL SEDIMENTS**
Aude Flamand[1] (ORCID : 0000-0001-9983-4176)
Jean-François Lapierre[2,3] (ORCID : 0000-0001-5862-7955)
Gwénaëlle Chaillou[1] (ORCID: 0000-0002-4170-8852)
[1] Québec-Océan, Institut des Sciences de la Mer de Rimouski, Université du Québec à Rimouski, 310 Allée des
Ursulines, Rimouski, Québec, Canada, G5L 3AL.
[2] Département de sciences biologiques, Université de Montréal, 1375 Avenue Thérèse-Lavoie-Roux, Montréal,
Québec, Canada, H2V 0B3.
[3] Groupe de Recherche Interuniversitaire en Limnologie (GRIL).
**Corresponding Author:** Gwénaëlle Chaillou. Email: gwenaelle_chaillou@uqar.ca



**ABSTRACT**
Increasing rates of coastal erosion and permafrost thaw along the Arctic coastline represent a major lateral
source of dissolved organic matter (DOM) to the coastal environment, where it can meet multiple fates
depending on its origin and composition. Along the (ground)water flow path, Iron (Fe)-hydroxides play an
important role in the retention of terrestrial organic matter, but its role on DOM released from coastal thawing
permafrost specifically remains poorly understood.  To address this gap, we sampled permafrost meltwater,
beach groundwater, and seawater samples from several coastal bluffs transects up to 2 km from the shoreline.
Across the salinity gradient – from permafrost meltwater to nearshore waters - we found that dissolved organic
carbon (DOC) and chromophoric dissolved organic matter (CDOM) concentrations decreased drastically,
indicating significant removal processes along this continuum. Optical indices (aCDOM350, SUVA254, HIX)
reflected changes in DOM composition and aromaticity, suggesting microbial degradation and mineral-organic
interactions occur to transform DOM. Furthermore, a PARAFAC analysis of fluorescent DOM indicated that
permafrost-derived DOM had a high molecular weight (HMW), humic, and terrigenous origin, while coastal
ocean-derived FDOM was protein-rich, low molecular weight (LMW), and from microbial (autochthonous)
origin. The optical signature of permafrost meltwater faded along the permafrost-nearshore water continuum.
Controlled experiments with excess $Fe^{2+}$ along constant oxygen bubbling showed a rapid (within 6 hours) and
major decrease in DOC and CDOM, suggesting interaction with reactive Fe-hydroxides, acting as a permanent
or temporary trap of permafrost-derived DOM. Overall, our findings highlight the role of intertidal and
nearshore zones where subsurface flows regulate the persistence and reactivity of terrestrial DOM as it transits
from permafrost to marine environments in the Arctic.
**KEYWORDS:** permafrost, Arctic coastal ocean, dissolved organic matter, dissolved organic carbon, iron
hydroxide.





**GRAPHICAL ABSTRACT**

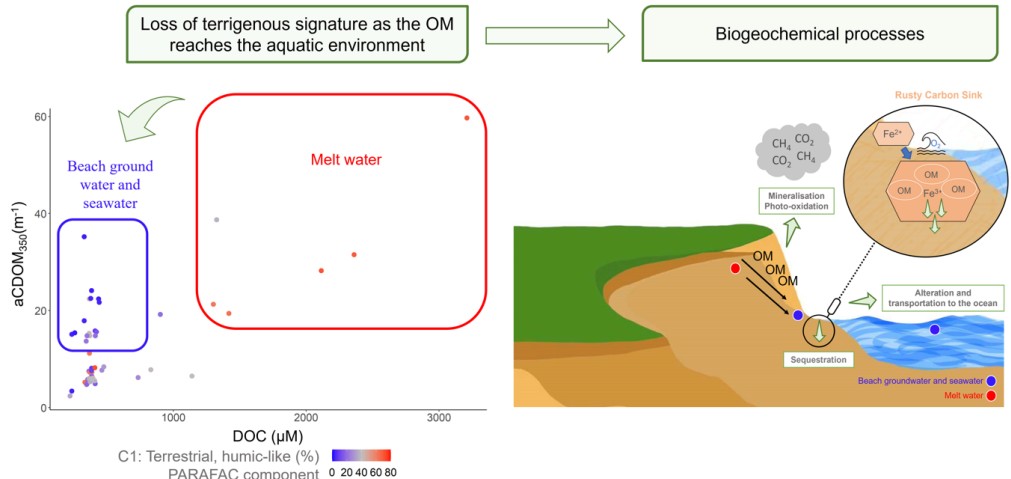






## 1 INTRODUCTION


Permafrost stores around 1,300 Pg of organic carbon (OC) within its $13.9 \times 10^6$ km$^2$ surface area, which
represents 60 % of the world's carbon stored in 15 % of the world's soil (Obu et al. 2019; Schuur et al. 2015).
The Arctic permafrost coastline is greatly impacted by global changes inducing unprecedented thawing rates,
along with the deepening of the active layer, which, in turn, increases subsurface transport (Jones et al. 2020;
Lantuit et al. 2012). Unlithified and ice-bonded permafrost cliffs, such as those that span the Beaufort Sea, are
susceptible to coastal erosion. Over the past twenty years, they have experienced one of the highest coastal
erosion rates recorded in the Arctic, with a recorded rate of 1.1 m yr$^{-1}$ between 1950 and 2000. This rate has
increased by 80–160 % in the last two decades (Jones et al. 2020; Lantuit et al. 2012). Accelerating coastline
erosion is supplying increasing quantities of terrestrial materials (Kipp et al. 2018), associated nutrients (Fritz
et al. 2017), carbon (Bristol et al. 2021), and contaminants (Kwasigroch et al. 2018) to the nearshore and coastal
ocean. This additional, non-point source of solutes is remobilized in late summer mostly when thaw depths are
at a maximum, and rapidly reaches nearshore waters via surficial and subsurface flows (Walvoord & Striegl
2007; Lecher, 2017).
Dissolved organic matter (DOM) represents a fundamental link between terrestrial and aquatic carbon cycles
and plays a significant role in the biogeochemistry of aquatic ecosystems (Hedges & Keil 1995). Terrestrially
derived dissolved organic matter (tDOM) strongly influences coastal ecosystem functioning (Vonk et al. 2015),
food web dynamics (McMeans et al. 2015; Thingstad et al. 2008), ocean chemistry (Guo et al. 2007; Stedmon
et al. 2011; Vonk et al. 2014) and optical conditions (Fichot et al. 2013; Matsuoka et al. 2012). A fraction of
this tDOM can be rapidly mineralized through microbial and photochemical processes, affecting nutrient
budgets, air-sea $CO_2$ exchanges, biological productivity, as well as acidification, in coastal waters (Kaiser et al.
2017a; Kaiser et al. 2017b). For example, Kaiser et al. (2017a) showed that ~50 % of the annual tDOC
discharged by Siberian rivers was mineralized along the land-sea continuum: tDOC is strongly removed and
lost as $CO_2$ along the transport. Therefore, only a small fraction potentially persists in the ocean over centuries
and millennia (Fichot & Benner 2014; Kaiser et al. 2017a). While the export of tDOC is known to strongly



influence the arctic marine ecosystem, little is known about the role and importance of erosional and thawing
inputs in shaping the ecology and chemistry of nearshore coastal waters. This is due not only to the stochastic
nature of erosion and thaw-related inputs but also to the complex nature of mineral and organic phases and their
role in tDOM stabilization along the flow path. The mechanisms and processes related to organic matter
transformations occurring at mineral-organic interfaces are complex (see Li et al., 2023; and references therein),
particularly in dynamic systems where biological, geochemical, and redox conditions interplay to influence the
concentrations and molecular compositions of DOM. For example, the formation of an iron (Fe) curtain can
represent an important mechanism of terrigenous OC storage (Zhou et al. 2024), particularly along subterranean
estuaries (STE) and intertidal discharge zones (Riedel et al., 2013; Linkhorst et al., 2017; Sirois et al. 2018;
Zhou et al., 2023). In temperate and subarctic regions, STE is a complex hydrogeochemical system along the
groundwater flow path which acts as a biogeochemical reactor where DOM is mineralized and/or trapped
(Anschutz et al., 2009; Sirois et al., 2018; Hébert et al., 2022). Its role as a transient or permanent terrestrial
organic carbon sink in the Arctic region is not known but could be a key zone of permafrost-derived DOM
trapping. The behaviour and optical properties of the permafrost-derived DOM as it reaches STE and nearshore
waters remain unclear, as does their affinity with amorphous Fe-hydroxide in STE.
To understand the behaviour of DOM along the land-sea continuum, absorbance- and fluorescence-derived
indices are commonly used to characterize its origin, reactivity, and transformations (Fichot & Benner 2014;
Meilleur et al. 2023; Stedmon et al. 2003). In addition, the use of excitation-emission matrices (EEM) with
parallel factor analysis (EEM-PARAFAC) of FDOM (Bro 1997) can allow for assessment of the composition
and sources of permafrost-derived DOM delivered to nearshore waters via surface runoff and groundwaters.
Recent findings by Fouché et al., (2020) characterized the permafrost-derived DOM as low molecular weight
(LMW), proteinaceous and with low aromaticity, with this signature fading rapidly during lateral flow
downslope of the permafrost table and within the fluvial continuum. This suggests that permafrost-derived
DOM may be rapidly lost in the permafrost – nearshore water continuum. However, further investigation is
needed to understand the mechanisms and processes that control this loss as DOM flows within intertidal
sediments and into the coastal Arctic Ocean. This study aims to characterize the transformation pathways of



DOM released from the thawing of coastal permafrost cliffs, while better understanding the role of beach
groundwater in the transfer of tDOM from permafrost to nearshore waters. More specifically, we have
developed a site-specific scale approach (<2 km) in the Kugmallit Bay (NWT, Canada) to optically characterize
and follow the behaviour of the DOC and DOM (CDOM and FDOM) along the permafrost – nearshore water
continuum. In addition, the affinity of permafrost-derived DOM and DOC with amorphous Fe-hydroxides, as
they flow from the subsurface to nearshore waters, was experimentally tested.

## 2  MATERIALS AND METHODS

### 2.1  Site Description

The study area is located in the Inuvialuit Settlement Region of the Northwest Territories adjacent to the
Mackenzie Delta region, the 4[th] largest river draining in the Arctic Ocean (Macdonald et al. 1998). A first
sampling campaign took place from July 24[th] to August 6[th], 2019, and a second campaign from June 22[nd] to
August 31[st], 2021, when thaw rates were at a maximum. About 60 samples were collected at four sampling sites
characterized by continuous permafrost coastal cliffs with thaw slumps surrounded by sandy and clay beaches:
Tuktoyaktuk Island, Peninsula Point, Crumbling Point and Reindeer Island (Fig. 1). Tuktoyaktuk Island, the
main sampling site (N=27), is characterized by a coastal bluff of approximately 9 m high, 1.5 km length and
100 m width (Ouellette 2021) and is located across the Hamlet of Tuktoyaktuk, in the south-east of the
Kugmallit Bay. The island loses ~1.8 m of shore per year due to erosion induced by storms and thawing
permafrost; an increase of 22% has been observed in the last 15 years (Berry et al. 2021; Tanguy et al. 2023;
Whalen et al. 2022), and the site is projected to entirely disappear within 20–30 years (Jones et al. 2020). In
front of the cliffs, a ~50 m wide beach is composed of a 0.3 to 0.5 m deep layer of fine to medium sandy
sediment that overlays a frozen clay horizon. Peninsula Point site is in the Pingo Canadian Landmark, southwest
of Tuktoyaktuk Island, and forms a complex retrogressive thaw slump system known for the presence of a
massive ice body of between 5 m and 20 m thickness (Mackay, 1986). Large muddy lobs composed of thawed
permafrost material and meltwater flow downslope to the nearshore (Hayes et al., 2022) where they sporadically



cover sandy intertidal sediment. Based on Hayes (2020), the recent shoreline retreat was of ~3.4 m yr$^{-1}$ from
1985 to 2018. Crumbling Point is also a retrogressive thaw slump system, located at the extreme northwest of
the Kugmallit Bay. Finally, Reindeer Island is located at the north of Richards Island, in an important lagoon
system formed by thermokarst lakes surrounded by coastal bluffs. To the best of our knowledge, there are no
published data on the coastal retreat in these zones, but it could be similar, at least, to what is reported in the
Canadian Beaufort-sea region (~0.5 m a$^{-1}$ (Solomon, 2005)) and likely reach very high local retreat rates as
presently observed in some location, as in Pullen Island (>12 m a$^{-1}$ (Berry et al. 2021)).
*2.2  Water and Sediment Sampling*
At each site, we carried out a site-specific scale sampling where different water sample types were collected
along a transect, from the coastal permafrost cliffs, through the sandy intertidal zones, to the near-shore
seawater. Meltwater and groundwater (here defined as porewater into fine sandy coastal sediment) samples
were collected on coastal permafrost slumps and the adjacent sandy shore, respectively. In contrast, seawater
samples were collected in front of each study site between 0.5 to 1 km from the coastline. Meltwater was directly
sampled in puddles formed on the slope of thaw slumps using a submersible pump. For beach groundwater,
push-point piezometers were inserted to ~50 cm depth into the sandy ground, above the frozen clay layer, in
front of thaw slumps in the intertidal zone and water was continuously pumped by a Solinst® peristaltic pump.
A massive ice sample was also collected from a permafrost core collected at Richard Island. The core was
sectionized and the different sections were defrosted gently in a hermetically closed acid-cleaned bucket. The
thawing water was collected by a peristaltic pump. Finally, seawater was collected in front of the slump systems
using a submersible pump placed between 0.5 and 1 m depth below the surface from a small vessel. For each
location, water samples were pumped into an online flow cell where practical salinity (S$p$), temperature and
oxygen saturation were monitored using a daily calibrated multiparametric probe (600QS, YSI Inc.).
After these parameters stabilized, water samples were collected for CDOM/FDOM into acid-washed 60 mL
glass amber bottles after on-line filtration through a 0.22 µm Millipore Opticap® XL4 cartridge with a
Durapore® membrane. The samples were stored in the dark at 4 °C. Total dissolved Fe samples were collected



in 60 mL metal-free Falcon® tubes after filtration through the same 0.22 µm Millipore Opticap cartridge. The
samples were acidified with 3 drops of 70 % nitric acid to prevent the re-oxidation of reduced trace metals and
stored at 4 °C. DOC samples were taken using 60 mL acid-cleaned polypropylene syringes and rapidly filtered
with pre-combusted (450 °C for 5–6 hours) 0.7 µm glass microfiber filters GF/F Whatman™ and stored in pre-
combusted and acid-washed 12 mL borosilicate EPA tubes with PTFE caps. The DOC samples were acidified
to pH <2 with high purity HCl 2N and stored in the dark at 4 °C until analysis. During the 2019 campaign,
samples were also collected in 30 ml scintillation vials, hermetically sealed for further water isotope analysis.
***2.3  Chemical and Optical Analysis***
Stable isotopes of water were analyzed by EA-IRMS during the following year after the collection. Accuracies
are ±0.05 ‰ and ±1 ‰ for $\delta^{18}O$ and $\delta^{2}H$, respectively. Reference materials were used throughout the isotopic
water analyses and isotopic analyses are reported compared to the international Vienna Standard Mean Ocean
Water (VSMOW). DOC samples were analyzed a few weeks after data collection by Total Organic Carbon
analyzer (TOC-Vcpn Shimadzu) based on the method of Wurl and Tsai (2009). The analytical uncertainty was
less than 4 %, while the detection limit was 5.8 µM. To ensure instrument stability, fresh acidified deionized
water (blank) and a standard solution (86.6 ± 1.7 µM) were regularly analyzed. The concentration of total
dissolved iron ($Fe_{tot}$) was measured according to the ferrozine method proposed by Stookey (1970) and adapted
by Viollier et al. (2000). The detection limit of the method was 0.4 µM and the reproducibility was better than

158     0.3 %.

Absorbance and fluorescence spectroscopy were used for the measurement of the chromophoric and fluorescent
fractions of DOM (CDOM and FDOM) a few weeks after sampling. The CDOM absorbance was measured
using a Lambda 850 UV-VIS Perkin Elmer spectrophotometer with 1 cm path-length quartz cuvettes.
Measurements were taken from 220 to 800 nm at 1 nm intervals with a scanning speed of 100 nm min$^{-1}$ and a
5 nm slit width. Blanks and references were measured using fresh Milli-Q water. The FDOM was measured
concomitantly using a Varian Cary Eclipse spectrofluorometer. Fluorescence spectra were measured within the
emission wavelengths of 220 to 600 nm and the excitation wavelengths of 220 to 450 nm at 5 nm intervals as



described by Couturier et al. (2016). Similarly, fresh Milli-Q water was used as a blank to rinse the cuvette in
between samples. Fresh deionized water was used as a blank and absorbance measurements of the samples were
used to correct the inner-filter effect and the dataset was corrected for Rayleigh and Raman scattering, according
to the method used by Pucher et al. (2019).
### *2.4 Optical-derived indices and PARAFAC model*
Absorbance and fluorescence indices were extracted using the staRdom toolbox on the R Studio Software
(Pucher et al. 2019). Different indices were explored but here we only reported 3 of them to characterize the
DOM pool because they presented significantly different values between the different categories of samples
(e.i. groundwater, melting water, massive ice, and seawater). The spectral absorption coefficient at 350 nm
(aCDOM350) was used as a tracer of CDOM absorption and content. It was calculated as 2.303 times the
absorbance at the wavelength $\lambda=350$ nm divided by the pathlength of the cuvette (m). The specific UV
absorbance (SUVA254 in mgC $L^{-1}$) was calculated as the absorbance at the wavelength $\lambda=254$ nm divided by
the DOC concentration. It allows tracking the CDOM aromaticity (Weishaar et al. 2003): greater $SUVA_{254}$
values correspond to a greater degree of aromaticity (Helms et al. 2008). It also has been shown as positively
correlated with the molecular weight of the DOM compounds.  In the FDOM pool, the humification index
(HIX) corresponds to the peak area under emission of 435–480 nm divided by the peak area under emission of
300–345 nm, at an excitation of 254 nm. HIX is an indicator of humic substances and the extent of humification
of DOM compounds (Hansen et al., 2016; Ohno, 2002): higher HIX indicates a greater humification of the
DOM source and HMW compounds.
In combination with the absorbance and fluorescence indices, a PARAFAC model was developed to investigate
further the composition and the sources of FDOM across samples (Bro 1997; Murphy et al. 2013; Stedmon et
al. 2003). Three components were validated using the method adapted from Pucher et al. (2019), in R studio
($R^2$>92 %). The components were also matched with the literature for identification and external validation,
using OpenFluor (Murphy et al. 2013). The three fluorescing peaks (C1-3) identified are presented in Fig. 2 and
their theoretical characteristics based on the literature are summarized in Table 1. Briefly, C1 and C3



components are mainly related to terrestrial and humic-like compounds of high (HMW) and low molecular
weight (LWM), respectively.  In contrast, the C2 component is likely related to freshly produced protein-like
compounds, of autochthonous origin and is mainly composed of LMW compounds.
*2.5  Affinity of Permafrost derived DOM with Iron Hydroxides*
Iron-spiked experiments were performed to assess the affinity of flowing DOM and DOC with amorphous Fe-
hydroxides. Samples of beach groundwater, seawater and meltwater were collected in 1-L acid-washed glass
bottles. In the laboratory, ~20 mM of $FeCl_2 \cdot 4H_2O$ were added to filtered (Pall® GWV High-Capacity
Groundwater Sampling Capsule, 0.45 µm porosity) water samples. The concentration of Fe was intentionally
added in excess compared to $Fe_{tot}$ concentration measured in the samples (median $Fe_{tot}$ concentration ~ 1.1 µmol
$L^{-1}$; with maximum values of 680 µmol $L^{-1}$) to favor the oxidative precipitation of amorphous Fe-hydroxides.
The experiments were performed rapidly after the sampling (<24h) during the 2021 campaign. The
experimental bottles were kept in the dark, at room temperature (~ 21°C). They were continuously air-bubbled
to maintain well-oxygenated conditions and favoured the precipitation of Fe-hydroxides. Sub-samples for DOC
and CDOM analysis were collected at times 0, 6, 12, 24 and 48 hours. DOC and CDOM samples were analyzed
as previously described.
**3    RESULTS AND DISCUSSION**
*3.1  Physico-chemical characteristics along the permafrost-nearshore continuum*
In this study, we refer to samples collected at the nearshore as "seawater"; nevertheless, we acknowledge that
these samples more accurately represent a brackish environment, with practical salinity S*p* values ranging from
0.5 to 20. The massive ice and meltwater samples exhibited the lowest salinities with S*p*<1 whereas the salinity
of beach groundwater samples ranged between 0 and 5.4. The S*p* range measured in these groundwater samples
mostly reflected the tidal pumping effect and the recirculation of the seawater within the permeable sediments.
The higher salinities (S*p*>5.4) were only measured in seawater samples. The temperature varied between 8.1
and 16.7 °C (with a mean value of 13.0 ± 2.6 °C) with the higher temperatures measured in meltwater and some



beach groundwater samples. Oxygen saturations ranged from 6 to 141 % of saturation. The nearshore surface
seawater and the meltwater samples were all over-saturated because of their contact with the atmosphere.
However, the low-salinity beach groundwater samples exhibited low oxygen saturation (6 - 48 %) despite the
recirculation of well-oxygenated seawater. Redox oscillations and transitory oxygen-depleted conditions are
observed in microtidal sandy intertidal zones (Hébert et al. 2022; Sirois et al. 2018; Waska et al. 2021) where
the tidally input of oxygen is rapidly consumed by heterotrophic processes (Chaillou et al., 2024; Moore et al.,

221  2024).

***3.2  Origin of the subsurface water flow in the intertidal zone***
The $\delta^{18}$O and $\delta^2$H values measured in water samples collected in 2019, mostly at Tuktoyaktuk Island, Peninsula
Point sites, and Crumbling Point (Fig. 1), ranged from -28 to -10 ‰ and from -215 to -82 ‰, respectively, the
massive ice sample (N=1) presenting the most depleted signature (Fig. 3). These depleted values are largely
explained by low air temperatures and are typical of permafrost hydrology reported in the western Arctic (Fritz
et al. 2011; Utting et al., 2012). The samples are well aligned along the local meteoric water line (LMWL; $\delta^2$H
= 7.39 × $\delta^{18}$O - 6.70; Fritz et al., 2022), whatever their salinity values, except for the three (3) meltwater samples
that are slightly below it, probably due to evaporation processes at the surface. The similarity between the
massive ice, beach groundwater and seawater isotopic distribution and the LMWL regression line suggests a
common meteoric origin, probably from permafrost watershed. The subsurface flow that transits across the
beach sediment does not seem to be affected by surficial processes (e.g. evaporation process), as observed in
the meltwaters, likely limiting photochemical degradation of the flowing DOM.  These results agree with recent
study of Kipp et al. (submitted) that showed the occurrence of high activities of radon isotope ($^{222}$Rn) in the
same groundwater samples, a noble gas that rapidly escapes as soon as it is in contact with the atmosphere. The
absence of light and the low oxygen content in the subsurface were then suitable for microbial transformations
and mineral-organic interactions as observed in other STEs, both limiting the export of tDOM into adjacent
coastal waters (Couturier et al., 2017; Linkhorst et al., 2017; Sirois et al. 2018; Hébert et al., 2022; Zhou et al.,

239  2023).





### 3.3 Behaviour of the DOC and DOM pool


The distribution of the variables used to characterize the DOM pool along the permafrost-nearshore water
continuum is presented in Fig. 4. The DOC concentrations dropped from 2,360 µmol L$^{-1}$ in meltwater samples
to 236 µmol L$^{-1}$ in the saltiest seawater sample (Fig. 4A). The concentrations decreased sharply along the
continuum to reach values lower than 400 µmol L$^{-1}$ in beach groundwater and seawater, whatever the salinity.
Absorption coefficients at 350 nm were less variable, from 2 to an extreme value of 134 m$^{-1}$ measured in one
meltwater sample (Fig. 4B). As for the DOC concentrations, the aCDOM350 decreased along the continuum,
with a median value of 24.0 m$^{-1}$ in meltwater samples and median values of 8.4 and 7.8 m$^{-1}$ in beach groundwater
and seawater, respectively. The HIX values exhibited a large range of values for each type of sample. The
median values, however, tended to decrease along the continuum, from 3.5 (unitless) in meltwater samples to
1.4 in beach groundwater and 0.8 in seawater samples (Fig. 4C). Whereas DOC, aCDOM350 and HIX values
negatively decreased along the continuum, the SUVA254 values tend to increase from the massive ice and
meltwater samples to the beach groundwater and nearshore seawater samples. The median SUVA254 value of
2.4 mg C L$^{-1}$ in the meltwater samples increased slightly to median values of 2.9 mg C L$^{-1}$ in the beach
groundwater samples and they reached a median value of 3.4 mg C L$^{-1}$ in seawater samples (Fig. 4D).
DOC concentration and aCDOM are routinely used as proxies to characterize the quantity and quality of the
DOM pool in aquatic continuum. The relationships in between is used to reveal the biogeochemical source and
processing of organic matter through physical and biogeochemical conditions (Massicotte et al., 2017; Fichot
and Benner, 2011; Spencer et al., 2013; Stedmon et al., 2003). A linear relationship between DOC and aCDOM
means that the DOC portion stays constant within the DOM pool, whatever the salinity and their respective
origin. In freshwater systems, for example, DOC concentrations were often highly correlated with the DOM
pool (Frenette et al., 2012; Massicotte et al., 2017 and reference therein). However, the decoupling between
DOC and aCDOM350 was observed as soon as mixing, photo-oxidation, and microbial degradation operate at
different rates on DOC and CDOM/FDOM fractions of the DOM pool (Del Vecchio and Blough, 2004; Nelson
et al., 1998; Nelson and Siegel, 2013). This decoupling suggests active processing of DOM during its transit
from freshwater to marine environment, for example, in subterranean estuaries in which the photo-oxidation



processes were null, the CDOM and DOC coupling resulted from microbial degradation that simultaneously
mineralized both (Hébert et al., 2022). In contrast, Couturier et al. (2016) showed a strong CDOM-DOC
decoupling in another STE, with most of the high molecular weight (HMW) DOM compounds tending to be
trapped in the system and not reaching the receiving nearshore waters. In this latter STE, Sirois et al. (2018)
highlighted the importance of the Fe curtain, where reactive Fe phases in sediments act as an efficient trap for
terrestrial DOM at the oxic/anoxic interface, thereby promoting its long-term sequestration. The exact
mechanisms of the Fe-DOM trapping in STEs are not well known.  However, Linkhorst et al. (2018) showed
that the precipitation of amorphous Fe-oxides preferentially traps HMW compounds enriched in aromatic,
carboxylic, and hydroxyl moieties, such as altered lignin and polysaccharide compounds of terrestrial origin,
compared to the more aliphatic-rich compounds characteristic of marine DOM.
The DOC-CDOM decoupling observed along the permafrost-nearshore continuum (Fig. 5) suggested the
occurrence of distinct transformative processes between DOC and CDOM in beach groundwater and nearshore
waters, irrespective of salinity values. In the intertidal zone, the mixing between $O_2$-depleted beach groundwater
and well-oxygenated seawater induced suitable conditions for oxidative precipitation of Fe. In the absence of
light, microbial degradation and mineral-organic interactions likely dominated the fate of the flowing DOM
pool. As these processes operate simultaneously, they tend to decrease DOC and CDOM concentrations along
the continuum and change the degree of humification, lowering the molecular weight of the flowing material
(Fig. 4A to 4C). Despite this general trend, the impact on the aromaticity, as revealed by the SUVA values, is
less significant and surprisingly, the degree of aromaticity of the material tends to slightly increase along the
continuum. This suggests a higher proportion of aromatic compounds in the DOM pool. This increase is likely
due to the selective degradation of larger organic molecules by microbial and photochemical processes taking
place in the fresh-to-saltwater continuum (Benner and Amon, 2015). This increase could also be explained by
the preferential precipitation of non-humic material. As a result, the remaining DOM pool becomes dominated
by smaller, more aromatic compounds, thereby enhancing the overall aromaticity of the DOM. The low
fluorescent and biological indexes (FI<1.5, 0.5<BIX<0.9; data not shown) in nearshore water samples in
addition to high SUVA values indicated the occurrence of decomposed and more refractory DOM (McKnight



et al. 2001; Huguet et al. 2009) probably resulting from the large draining of the Mackenzie River. However,
the magnitude of the production of autochthonous DOM might be equivalent to the degradation processes,
which explains why the optical parameters in nearshore waters remain stable, regardless of the salinity. Indeed,
it suggests that freshly produced, protein-like FDOM (C2) from permafrost meltwater could be replaced by an
equivalent amount from local autochthonous sources, maintaining a consistent overall contribution of this DOM
despite variations in source origins.
### 3.4 Affinity with amorphous Fe-hydroxides and DOC-DOM decoupling
The different affinity of DOC and CDOM350 on Fe-hydroxides was experimentally tested by carrying out Fe-
spiked experiments of filtered meltwater, groundwater, and nearshore seawater samples. Here, the Fe-spiked
experiments were carried out to promote oxidative precipitation of amorphous Fe-hydroxides irrespective of
the total dissolved Fe concentrations measured in our samples (between the limit of detection to 680 $\mu$mol $L^{-1}$
in some groundwater samples). The sporadic presence of high $Fe_{tot}$ concentration in beach groundwater samples
likely results from the redox oscillation tidally induced by the input of well-oxygenated seawater as currently
observed in STE systems (Charette et al., 2002, 2006).
In the Fe-spiked experiments, the initial DOC concentration of the seawater, groundwater and meltwater
samples were 383, 334 and 1019 $\mu$mol $L^{-1}$, respectively, in agreement with the median DOC values reported in
Fig. 4A for the different types of samples. As soon as Fe was added, the DOC concentrations dropped rapidly,
losing ~40% of initial concentrations. Then, the DOC concentrations decreased over the next hours, reaching
their lowest concentrations 6 hours after the start of the incubation (Fig. 6A). After 48 hours, however, the DOC
was gradually released in solution to reach a final concentration of 260, 219 and 710 $\mu$mol $L^{-1}$ for nearshore
seawater, beach groundwater, and meltwater samples, respectively. Since the experiment was conducted for
only 48 hours, it is uncertain whether DOC concentrations continued to increase beyond this point or if a plateau
was eventually reached. Over the 48h experiment, however, the DOC seemed to be gradually desorbed from
the Fe-mineral phase and the net loss of DOC in the solution was only 32%, 34% and 30% for nearshore
seawaters, beach groundwaters and meltwaters, respectively. The Fe-DOC trapping showed consistent



behaviour in all three sample types, indicating that the DOC pool reacted in the same way regardless of salinity.
In contrast, the loss of aCDOM350 (or tDOM) in the solution was almost complete 6 hours after the Fe-spike
and the concentrations remained very low over the rest of the experiment (Fig 6B). At the end of the experiment,
the net tDOM loss was 62%, 57% and 94% of the initial content of nearshore seawaters, beach groundwaters
and meltwaters, respectively. The meltwaters exhibited the strongest tDOM loss in agreement with the initial
occurrence of HMW compounds with a high degree of humification, a material likely stabilized by amorphous
Fe.  The preferential trapping of specific compounds during the transit favours the export of non-Fe-stabilized
material from beach groundwaters to nearshore seawaters. The DOC-CDOM decoupling observed along the
continuum might thus result from their different affinities on the amorphous Fe-mineral surface. The exact
mechanism controlling the molecular fractionation of the DOM pool in arctic groundwater remains to be
determined, and further studies are required to explore the role of Fe-curtain in arctic STEs.
### 3.5  PARAFAC components in the DOM pool
Among the three fluorescing peaks identified by the PARAFAC model, the C2 component mostly dominated
the FDOM pool whatever the type of samples and salinities (Fig.7). No significant trend in component
distribution was observed along the salinity gradient. The protein-like compound C2 was significantly
negatively correlated to C1 and C3 ($r^2$=-0.89, p <0.001) and HIX ($r^2$=-0.92, p <0.001) and the median values of
C2 increased along the continuum, the highest median value being in the nearshore seawater samples. The two
humic-like components (C1 and C3) were well correlated with each other ($r^2$=0.81, p <0.001) and with HIX
($r^2$=0.85 and 0.69, respectively, p <0.001). As observed for HIX, DOC and aCDOM350, C1 and C3 decreased
along the continuum, with the lowest median values being measured in the seawater samples.
Upstream of the permafrost-nearshore continuum, massive ice and meltwater samples were mostly composed
of humic-like, HMW and terrestrially derived FDOM (C1) and they are rich in DOM and DOC, agreeing with
the active layer-derived FDOM also measured by Fouché et al. (2020). As the DOM transits, the loss of humic-
like compounds appears concomitant to the production of biologically-derived tyrosine-like FDOM, which is
typically produced by microbial organisms (Table 1 and references therein). The occurrence of non-Fe-



stabilized DOM and the redox conditions are then suitable for bacterial mineralization and the production of
lower MW and protein-like material. Bacterial mineralization of the transiting DOM is supported by the high
DIC concentrations measured in beach groundwater with concentrations higher than 3,000 µmol L$^{-1}$ as observed
in samples collected in 2019 (Lizotte et al, 2022). Subterranean estuaries are biogeochemical reactors where
solutes of both marine and terrestrial origin are transformed and released to nearshore waters (Moore, 1999;
Anschutz et al., 2009). Moreover, a recent study in a subarctic beach suggested that the discharge zone may be
a hot spot of $CO_2$ degassing (Chaillou et al., 2024). The fraction of the permafrost-derived DOM which escaped
the Fe-curtain along the groundwater pathway was highly transformed through microbial degradation before
becoming diluted with the marine DOM pool.
**4   CONCLUSION**
In this study, we observed a rapid decrease in DOC and CDOM concentration across a short spatial scale,
indicating rapid and significant removal processes as DOM flows across the land-nearshore water continuum.
Microbial degradation and mineral-organic interactions would be preferentially removing HMW humic-like
material, leaving behind more aromatic, refractory compounds. Fe-hydroxides appear to play a key role in
rapidly and selectively trapping this tDOM during subsurface water transit, acting as a sink and shaping the
composition and concentration of DOM released in nearshore waters. The contribution of beach groundwater
and associated submarine discharge at the front of the coastal bluffs remains to be quantified, as it may regulate
carbon exports from permafrost watersheds to the Arctic Coastal Ocean.
Our findings highlight the role of intertidal and nearshore zones in regulating the persistence and reactivity of
terrestrial DOM as it transits from terrestrial to marine environments. The rapid loss of permafrost-derived
DOM in these environments, coupled with its interaction with mineral phases like amorphous iron oxides,
suggests that these zones may act as a permanent or transient terrestrial carbon sinks, as also observed in
temperate regions. However, the potential for rapid transformation and mineralization of this carbon along the
land-sea continuum indicates that much of it may be lost as $CO_2$ before reaching the ocean. This study



underscores the need for further research to understand the fate of DOM in Arctic coastal regions, particularly
in the context of accelerating permafrost thaw and coastal erosion. Further research is crucial for predicting the
impact of Arctic carbon fluxes on global biogeochemical cycles and developing strategies to mitigate the
consequences of permafrost degradation on climate systems. Given the ongoing effects of climate change, there
is an urgent need to comprehensively characterize and quantify these lateral and non-point source of carbon
within coastal Arctic budgets.
**Data availability:** Along with this submission, the dataset used in this research was submitted and accepted for
publication to Pangaea Data Publisher (www.pangaea.de). Once this article accepted for publication, the
moratorium in place will be lifted and the dataset generated during the study will be freely available in the
Pangaea repository. Here is the hyperlink and DOI toward the dataset:
https://doi.pangaea.de/10.1594/PANGAEA.960986
**Author contribution:** Aude Flamand: Investigation, Conceptualization, Methodology, Formal analysis,
Validation, Data curation, Visualization, Writing - Original Draft Preparation, Writing - Review & Editing;
Jean-François Lapierre: Supervision, Methodology, Validation, Data curation, Writing - Review & Editing;
Gwénaëlle Chaillou: Conceptualization, Methodology, Validation, Data curation, Supervision, Project
administration, Funding acquisition, Writing - Review & Editing.
**Competing interests:** The authors declare that there are no competing interests.
**Acknowledgements:** This project was made possible through the tremendous support of the hunters and
trappers committee, the hamlet and town council, in addition to several members of the Inuvialuit Nunangit
Sannaiqtuaq community of Tuktoyaktuk. The authors would like to express their gratitude to Dustin Whalen
(NRCan) for his invaluable contributions that made the fieldwork possible, as well as for his assistance in
providing resources and coordinating the fieldwork. We also want to thank Charlotte Irish, James Pokiak, Angus
Robertson, Bay Berry, and Brian Mayhew for their precious help during fieldwork in the summers of 2019 and
2021. We thank Claude Belzile for analyzing the DOC sample at ISMER-UQAR, Frédérik Bélanger for
conducting the CDOM and FDOM analysis of the 2019 samples and Antoine Biehler for his help with database



management. We also want to thank Simon Bélanger (UQAR) and Celine Guéguen (U. Sherbrooke) for
providing early feedback on this manuscript. This work represents a contribution to the scientific programs of
Nunataryuk, ArcticNet, and Québec-Océan.
**Financial support:** Financial support for this project has been provided by the Network of Centers of
Excellence of Canada ArcticNet (grant no. P-66), by Québec-Océan, funded through the Fonds de Recherche
du Québec – Nature et Technologies, and by the Aurora Research Institute – Aurora College. The authors
received additional funding from NSERC (RGPIN-2021-04332 to GC) and in-kind support from Natural
Resources Canada (Climate Change Geoscience Program and Polar Continental Shelf Program; grant no. 007-
19) for logistics support (equipment and helicopter time), and Crown–Indigenous Relations and Northern
Affairs Canada (Climate Change Preparedness in the North program, CCPN grant no. CCPN PN-NT-077-2018;
Beaufort Sea Regional Strategic Environmental Assessment Program, BRSEA agreement no. 239) in the form
of consultation tour of the ISR (March 2019). AF also received grants from the Northern Scientific Training
Program (NSTP) and from Québec-Océan.



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



**TABLES**
Table 1: Description of the EEM-PARAFAC modelled FDOM components based on the literature results of
literature references. PARAFAC components and their characteristics

| Comp. | Peak max Ex/Em | Coble peak | Description | Literature |
|---|---|---|---|---|
| C1 | 250-335/466 | A, C | Humic-like terrestrial, HMW, aromatic | $C_C$: <240-340/452 (Olefeldt et al., 2014)<br>ALL1: 250-350/459 (Pitta and Zeri 2021)<br>C3: <240-355/476 (Stedmon and Markager, 2005a)<br>C3: 260-370/490 (Murphy et al., 2018)<br>C4: <255-360/460 (Fouché et al., 2020) |
| C2 | 265/296 | B | Protein-like, tyrosine, biological, microbial autochthonous origin. LMW phenolic compounds. | $C_{Ty}$: 270/<300 (Olefeldt et al., 2014)<br>ACT-10 C3: 270/302 (D'Andrilli and McConnell 2021)<br>C6: 280/338 (Stedmon and Markager, 2005a)<br>C1:275/<300 (Murphy et al. 2008)<br>C1: 275/306 (Fouché et al., 2020) |
| C3 | 250-295/414 | A, M | Humic-like, terrestrial, autochtonous production and microbial processing, LMW. | $C_M$:<240,305/404 (Olefeldt, Persson et al. 2014)<br>C2: <300/396 (Søndergaard, Stedmon et al. 2003)<br>C2 : 315/418 (Murphy et al., 2008)<br>C2: 310/ 415 (Fouché et al., 2020) |












**FIGURE CAPTIONS**
**Fig. 1** Map of the four sampling sites (red dots) located in the Northwest Territories, Canada.
**Fig. 2** EEMs of the 3-components PARAFAC model. Fluorescence is expressed in Raman Unit (R.U.).
**Fig. 3** Isotopic composition of massive ice, meltwater, beach groundwater and seawater samples collected in
2019 showing the mixing line between samples across the salinity gradient, from the meltwater to the
seawater. global (GMWL; Craig, 1961) and the local meteoric water line for Inuvik (LMWL, Fritz et al.,
2022) are also reported. Note that only one massive ice sample (N=1) was collected.
**Fig. 4** Distribution of (A) DOC, (B) CDOM350, (C) HIX and (D) SUVA254 indexes in the salinity gradient
and for the different types of collected samples (*i.e.,* beach groundwater, massive ice, meltwater and
nearshore seawater samples). Note that there is only one massive ice sample reported here.  For the boxplots,
the black lines are the median values, the whiskers are the extent of the data, and the dot points are the outlier
values.
**Fig. 5** Global relationship between absorption coefficients at 350 nm (aCDOM350 in m$^{-1}$) and DOC
concentrations (in µmol L$^{-1}$) along the permafrost to nearshore aquatic continuum. Note that the data is
reported in Log units.
**Fig. 6** Behaviour of (A) DOC and (B) aCDOM350 concentrations with excess iron and constant oxygenation
in the different type of samples incubated over 48 hours. The non-colored points represent the concentrations
before the addition of Fe-spike at t=0h.
**Fig. 7** Distribution of PARAFAC components (A) C1, (B) C2, and (C) C3 along the salinity gradient and for
the different types of collected samples (*i.e.,* beach groundwater, massive ice, meltwater and nearshore
seawater samples). Note that there is only one massive ice sample reported here.  For the boxplots, the black





lines represent the median values, the whiskers represent the extent of the data, and the dot points represent
the outlier values.



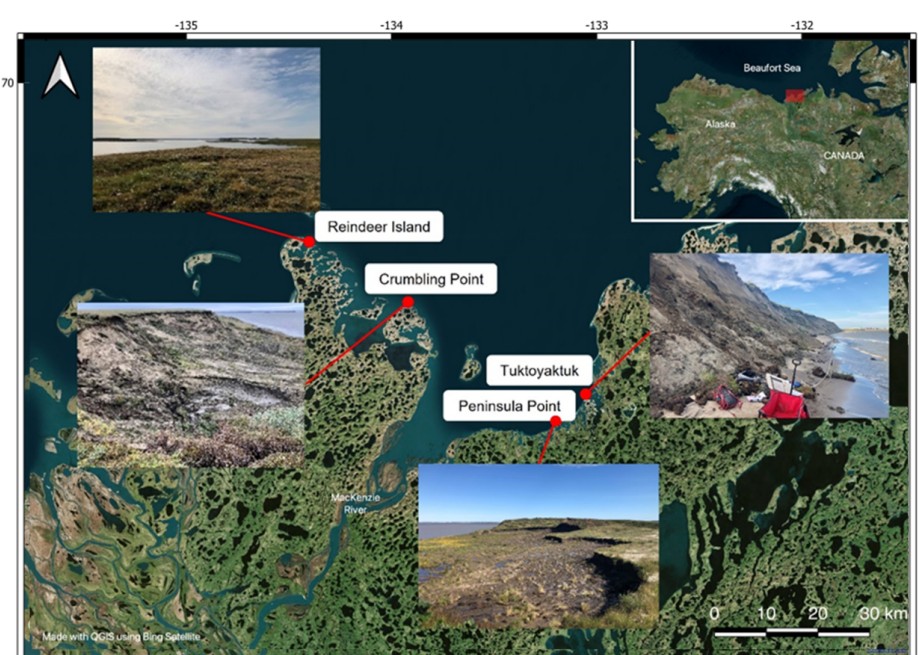


**Fig. 1** Map of the four sampling sites (red dots) located in the Northwest Territories, Canada.

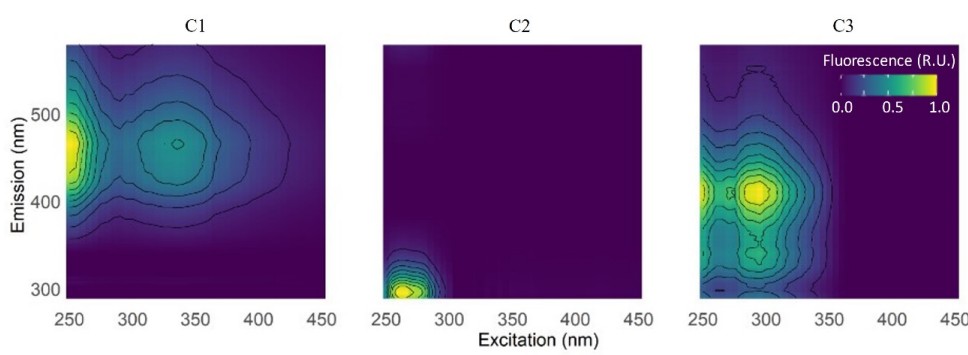


**Fig. 2** EEMs of the 3-components PARAFAC model. Fluorescence is expressed in Raman Unit (R.U.).


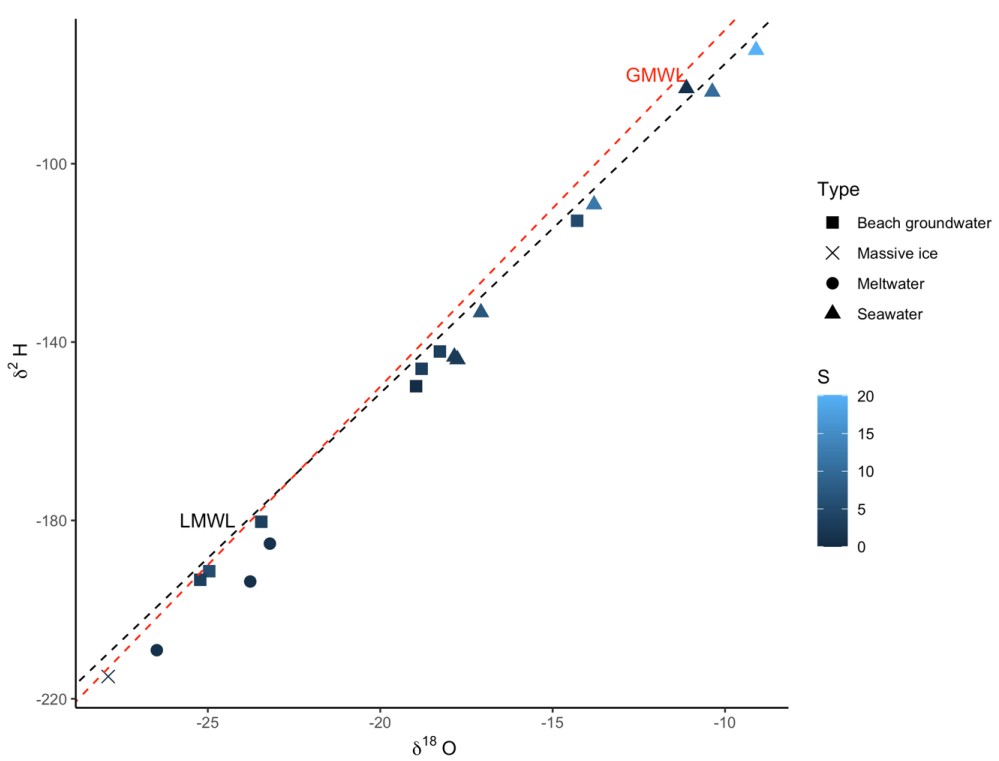

**Fig. 3** Isotopic composition of massive ice, meltwater, beach groundwater and seawater samples collected in 2019 showing the mixing line between samples across the salinity gradient, from the meltwater to the seawater. global (GMWL; Craig, 1961) and the local meteoric water line for Inuvik (LMWL, Fritz et al., 2022) are also reported. Note that only one massive ice sample (N=1) was collected.



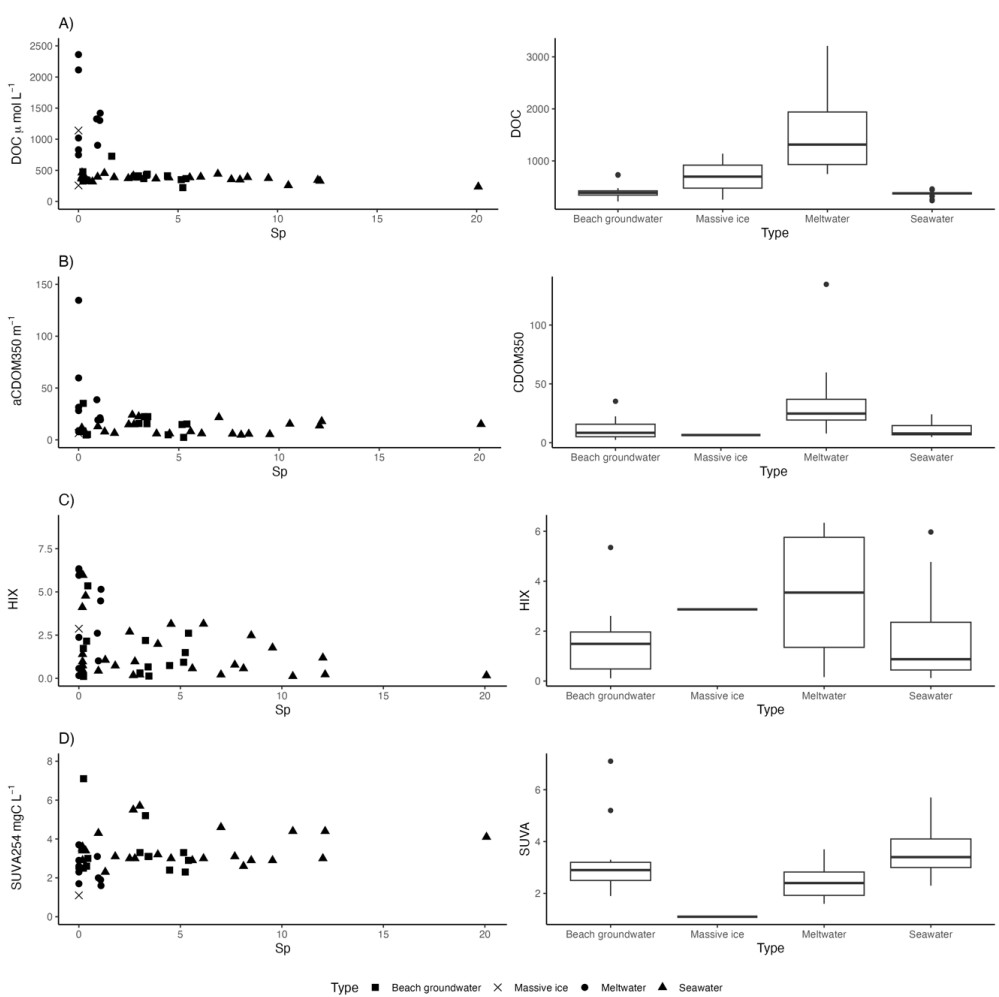

681

**Fig. 4** Distribution of (A) DOC, (B) CDOM350, (C) HIX and (D) SUVA254 indexes in the salinity gradient

and for the different types of collected samples (*i.e.,* beach groundwater, massive ice, meltwater and

nearshore seawater samples). Note that there is only one massive ice sample reported here. For the boxplots,

the black lines are the median values, the whiskers are the extent of the data, and the dot points are the outlier

values.




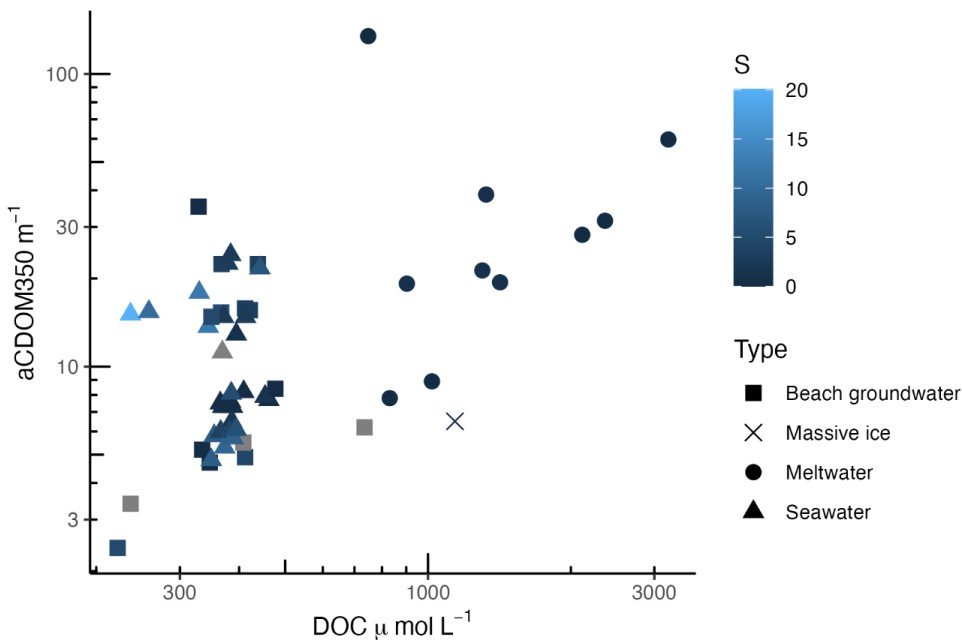


**Fig. 5** Global relationship between absorption coefficients at 350 nm (aCDOM350 in m$^{-1}$) and DOC

concentrations (in µmol L$^{-1}$) along the permafrost to nearshore aquatic continuum. Note that the data is

reported in Log units.






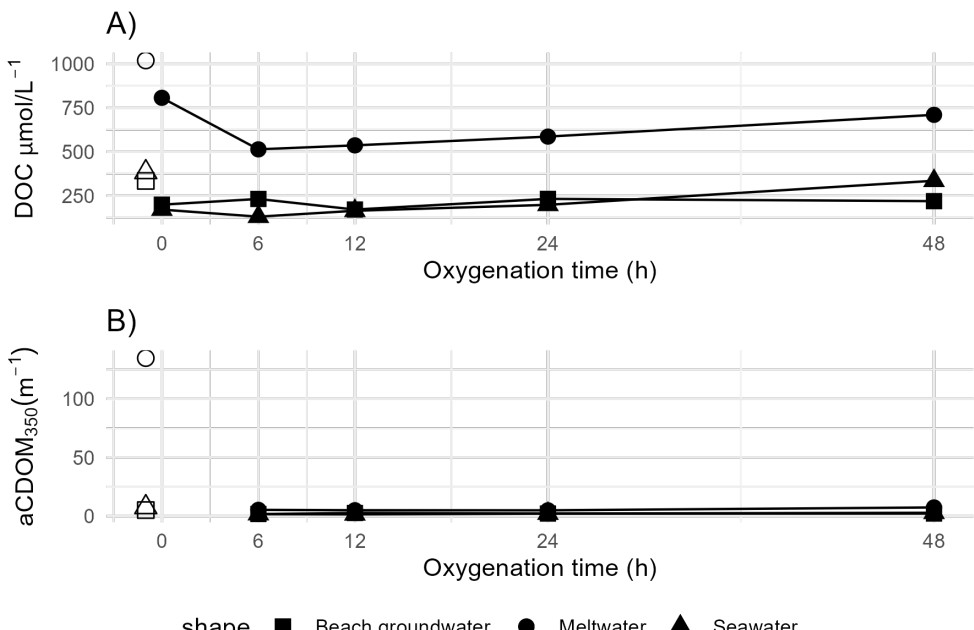


**Fig. 6** Behaviour of (A) DOC and (B) aCDOM350 concentrations with excess iron and constant oxygenation

in the different type of samples incubated over 48 hours. The non-colored points represent the concentrations

before the addition of Fe-spike at t=0h.




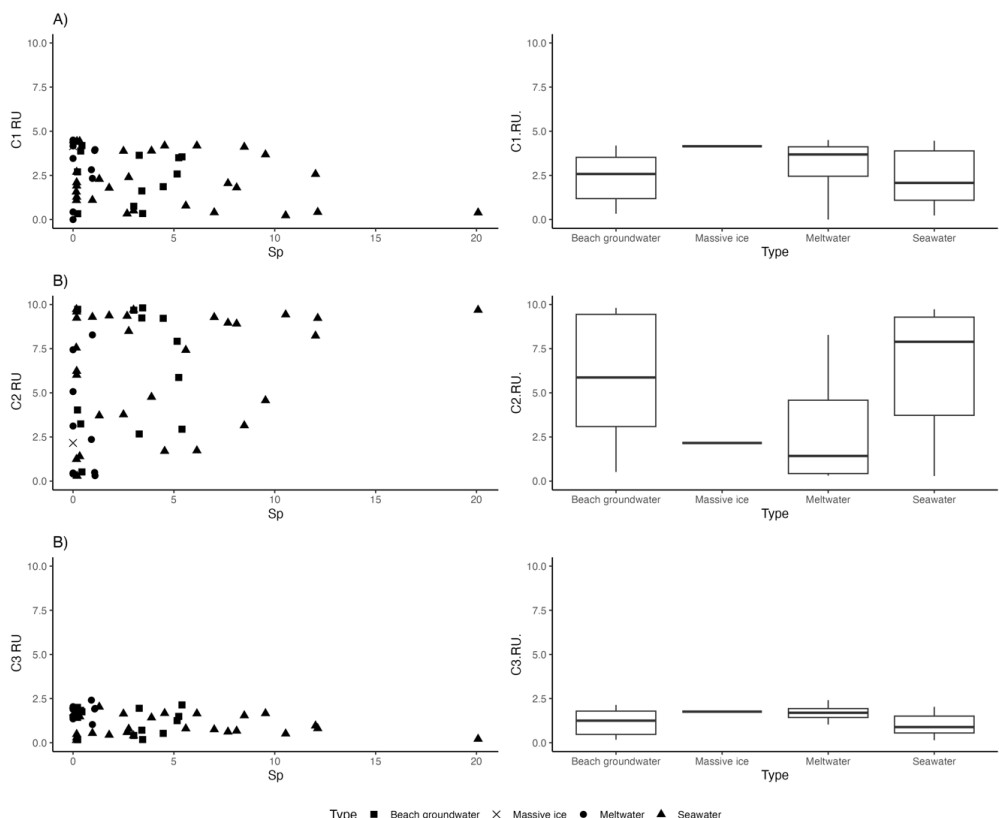


**Fig. 7** Distribution of PARAFAC components (A) C1, (B) C2, and (C) C3 along the salinity gradient and for
the different types of collected samples (*i.e.,* beach groundwater, massive ice, meltwater and nearshore
seawater samples). Note that there is only one massive ice sample reported here. For the boxplots, the black
lines represent the median values, the whiskers represent the extent of the data, and the dot points represent
the outlier values.

705

706

707