# Peer review of "FATE OF DISSOLVED ORGANIC MATTER ACROSS THE PERMAFROST-NEARSHORE WATER"

_EGUsphere, 2024_

## Author Comment (AC2)

**Editor #1:**

This is an interesting manuscript about a potentially important mechanism of C-sequestration or C-metabolization associated with permafrost thaw at coastlines. The authors explore the properties and concentrations of dissolved organic matter (DOM) sourced from thawing permafrost along Arctic coastlines. They set off with the idea that DOM mobilized by thawing reaches the Arctic Ocean through a coastal beach area, where sub-surface interactions with microbes or Fe could affect the fate of DOM. Their study largely rests on concentrations (DOC) and optical properties (absorbance, fluorescence) of DOM along a gradient from thawing permafrost to the coastal ocean, which in their specific area of research is heavily influenced by the Mackenzie River. In addition, they present experimental evidence that interactions with Fe could drive precipitation of DOM, thus representing a potential mechanism for C-sequestration in intertidal sediments.

The study is obviously of relevance in times of climate change with rapidly retreating Arctic coastlines caused by permafrost thawing. However, I find the study´s conclusions to be largely speculative and not resting on convincing evidence. In particular, I see three shortcomings that should be addressed before the manuscript may be considered for publication:

**The authors extend their gratitude to the editor for their incisive yet highly constructive comments. We have addressed most of the suggestions made by Referee 1 and the editor, and we believe that the manuscript has been significantly improved. Substantial revisions have been made to the introduction, and we will add a conceptual figure (see the following figure 1) to clearly illustrate our sampling process approach. Additionally, we conducted a more in-depth analysis of the conservative versus non-conservative behavior of DOC and DOM. Further data have been included in the form of tables and Figures (e.i. total dissolved iron, Fe/DOC, pH). However, since the full dataset is publicly available through PANGAEA platform, we chose not to include it in the supplementary information. We hope these revisions meet the requirements for publication.**

[Figure]

Figure 1 : Schematic cross-sections and photographs illustrating the permafrost–seawater continuum landscape at Tuktoyaktuk Island and Peninsula Point (not to scale). Black arrows indicate surface (solid lines) and subsurface (dashed lines) flow paths, while blue arrows represent seawater recirculation driven by tidal inputs into the intertidal zones of sandy beaches.

**Major comments:**

**COMMENT:**

The presented data largely consist of concentration gradients (DOC and other more or less quantitatively robust optical indicators) along a spatial gradient from thawing permafrost to the "open" coastal ocean. The latter is a brackish environment heavily influenced by the Mackenzie River. The authors observe a decrease of concentrations or optical signatures along this spatial gradient and along a gradient of increasing salinity. The offered interpretation is one tied to loss of DOM along this gradient, either through microbial respiration or through co-precipitation with Fe in intertidal sediments. The main issue I see is that the observed decrease of concentration may simply be caused by mixing of at least two different water bodies, namely thawing permafrost and seawater. If tidal influence (currently entirely unclear, no auxiliary data presented) plays a role, then there may even be a third water body, open ocean vs. river water, to consider. To conclude on carbon transformations from concentration gradients requires a careful consideration of mixing of various

water bodies along the main environmental gradient of the study. A worthwhile approach to explore could be to predict DOM gradients based on salinity and then more carefully explore differences to actually observed data.

**RESPONSE:**

We fully agree with this comment and acknowledge that we skipped important explanatory steps in the original manuscript. In the revised version, we will develop theoretical mixing lines using not only stable isotope data but also practical salinity values from the various samples. These theoretical mixing lines will be added to the figures, as illustrated in the updated version of Figure 2 below.

To better visualize our "seawater" and "beach groundwater" data distribution along the Sp gradient, we initially excluded samples from massive ice and meltwater. Based on the most enriched and most depleted stable isotope values within this dataset—corresponding respectively to samples with practical salinity values of 20.00 (N=1) and 0.18/0.30 (N=8)—we determined the average DOC and aCDOM$_{350}$ values for each mixing endmember. By connecting these two endpoints, we defined a theoretical mixing line, thereby assuming a conservative mixing behavior as a function of salinity.

While DOC concentrations from seawater samples—and to a lesser extent, from groundwater samples—align well along the theoretical mixing line, the aCDOM$_{350}$ data are more scattered, falling both above and below this line. This contrasting behavior between DOC and CDOM has been widely discussed in coastal environments (see Massicotte's review, Massicotte et al., 2017) and, to a lesser extent, in subterranean estuaries. The DOC-aCDOM relationship is often used to identify the biogeochemical processes (e.g., mixing, photo-oxidation, and microbial degradation) that regulate its conservative or non-conservative behavior in aquatic systems.Futhermoe, understanding and integrating the transformations of DOC and CDOM in these latter systems is a crucial step toward accurately quantifying carbon fluxes.

In this new version, we will clarify the behaviour of DOC and better integrate the role of dilution to explain the DOC distribution from beach groundwater to the seawater (open ocean water). However, the DOC/DOM concentrations measured in meltwater and massive ice can be explained by this process.

[Figure]

Figure 1: Distribution of DOC and aCDOM$_{350}$ vs. practical salinity (Sp) for "seawater" (blue dots) and "beach groundwater" samples (orange dots). A mixing line (in red) links the two theoretical endmembers. See the text for their choice.

**COMMENT:**

The authors speculate about microbial respiration in the sediments to potentially transform DOM from permafrost into CO2. There seems to be anecdotal evidence of high CO2-concentrations in intertidal sediments, but to really turn this into a strong argument more in-depth evidence for microbial respiration needs to be presented more carefully.

**RESPONSE:**

Unfortunately, DIC and $CH_4$ concentrations were only measured during the 2019 campaign, primarily due to the logistical challenges of preserving samples with $HgCl_2$ and transporting them to the south —challenges that were further compounded during the COVID-19 pandemic. We only have 5 samples. It therefore seems inappropriate to integrate these data. However, the presence of DIC (2800 – 4844 umol/l) and methane (41 to 297 nmol/l), along with $O_2$ undersaturation (>60%) and elevated DOC concentrations, strongly suggests that both aerobic and anaerobic mineralization processes are occurring, consistent with observations from subterranean estuaries in temperate regions. As the dilution cannot explain the DOM, and to a lesser extent DOC distribution in beach groundwater, other DOM-specific processes must be invoked to explain the discrepancy between DOC and DOM. In the new version, we will clarify this point.

**COMMENTS:**

The second speculated mechanisms for DOM loss along the studied gradient is an "iron curtain" associated with redox gradients in the sediment that could act to sequester carbon in beach sediments. The authors present results of an experiment that shows loss of DOM in an experimental setting with surplus Fe and under strong oxygenation. Long-term effects remain unclear, which is a bit worrisome as DOM seems to increase again towards the end of the experiment. The study does not present any evidence for Fe nor for any "iron curtain" to actually occur in the studied intertidal sediments. Last, while microbial respiration may be argued to be an ongoing transformation process capable of removing substantial amounts of DOM along its way from permafrost to the ocean, the iron curtain mechanism is likely limited by the amount of Fe available in the sediments. Is such an iron curtain mechanism actually capable of influencing DOM-delivery to the ocean to a quantitatively relevant amount? How much Fe is available? How much DOM needs to be sequestered by that mechanism to explain the concentration drop in intertidal sediments? There may be a chance to combine experimental data with in-situ Fe and DOM concentrations (or DOM fractions) in a few simple back-of-the envelope computations probing the iron curtain for its potential relevance?

**RESPONSE:**

These are indeed excellent questions that the scientific community is actively working to address, and we readily admit that our team does not yet have definitive answers, particularly for Arctic sandy beaches. The role of the so-called "iron curtain" in nearshore sediments—and, more broadly, organo-mineral interactions in trapping and mobilizing DOM—remains poorly understood. Redox conditions within STEs strongly influence the speciation, mobility, and fate of numerous chemicals, including ammonium, nitrate, dissolved manganese, and iron.

DOM has a strong affinity for freshly precipitated $Fe^{3+}$ (oxy)hydroxides at oxic-anoxic interfaces in intertidal sands, leading to the formation of Fe–organic matter (Fe–OM) associations. These complexes can temporarily stabilize organic matter against microbial remineralization (Linkhorst et al., 2017; Sirois et al.,

2018; Zhou et al., 2023). The extent of Fe–OM coprecipitation or coagulation is governed by several factors, including the DOC:Fe molar ratio, organic matter composition, Fe speciation, and physicochemical parameters such as pH and salinity. The chemical nature of DOM plays a critical role in its interaction with mineral surfaces: higher-molecular-weight, aromatic, oxygen-rich terrigenous DOM tends to bind more strongly to $Fe^{3+}$ than aliphatic, marine-derived DOM (Linkhorst et al., 2017). Additionally, slightly acidic conditions (pH 4–5) enhance DOM–$Fe^{3+}$ coagulation (Nierop et al., 2002; Amoako et al., 2025). Such organo-mineral associations influence both the bioavailability and transport dynamics of DOM and Fe, ultimately affecting their biogeochemical cycling and delivery to coastal waters (Linkhorst et al., 2017; Sirois et al., 2018). We will add this information in the introduction and use it more comprehensively in the discussion.

In reviewing the manuscript, we recognize that the experimental results may have been overemphasized, despite their speculative nature. In the revised version, we will retain the experimental details in the Methods and Results sections, but strive to integrate these results more carefully into the Discussion, using them to support our arguments rather than to dominate them.

In addition, we propose to incorporate complementary data on the particulate phase, including total organic carbon (TOC) content, as suggested by Referee 1, and reactive sedimentary iron, extracted using the citrate-dithionite-bicarbonate (CDB) reduction method of Mehra and Jackson (1960), as modified by Lalonde et al. (2012), and previously applied to sandy sediments by Sirois et al. (2018). These measurements were conducted on intertidal beach sediments collected during the 2019 field campaign at Tuktoyaktuk Island (N = 3) and Peninsula Point (N = 2). While we acknowledge that this limited dataset does not capture the full spatial variability of the system, it nonetheless supports the presence of iron oxide-coated sands in the region—a finding consistent with the expected influence of permafrost thaw as a source of iron. A new subsection will be added to the Methods to describe the particulate phase analyses (TOC and reactive Fe hydroxides; see Referee 1's comment), and a corresponding table (see below) will be included to present these data, which will inform part of the discussion.

Table 1 : Particulate fraction characteristics of sandy beach sediment collected at Tuktoyaktuk Island and Peninsula Point.

|  | Tuktoyaktuk Island N=3 | Peninsula Point N=2 |
| --- | --- | --- |
| Sediment depth layer (cm) | 0-50 | 0-30 |
| Total organic carbon (% dry sediment) | $0.43 \pm 0.12$ | $0.21 \pm 0.07$ |
| Reactive sediment Fe (ppm; µg Fe/g dry sediment) | $2{,}246 \pm 0{,}850$ | $1{,}320 \pm 0{,}250$ |

I have a few further minor comments, presented line-by-line below, that may help to provide an improved version of the manuscript.

**Minor comments:**

**Line 17:** "hydroxides" is plural, "its" is singular.

**Response:** Thank you for pointing that out, it will be corrected.

**Line 39:** Graphical abstract: The left side is of limited use, but the right side of the graph could be turned into a more useful figure showing study design and main speculated mechanisms. Should then become a main figure in the manuscript.

**Response:** We received a similar comment from Referee 1 and agree that the right side of the graphic has the potential to become a more useful figure. Please see the above comment and Figure 1.

**Lines 52-53:** a "source" cannot be mobilized, subject unclear in second sentence half.

**Response:** Thank you for pointing that out. The sentence can be modified to read: "This additional, non-point source releases solutes that are remobilized in late summer, when thaw depths are at their maximum. These solutes then rapidly reach nearshore waters via both surficial and subsurface flows."

**Line 74:** establish STE as abbreviation for singular, eventually then write STEs in case plural is needed.

**Response:** Thank you for the suggestion. We will define 'subterranean estuary' (STE) and use 'STEs' in the plural form where needed.

**Line 78:** subject unclear.

**Response:** We have received a similar comment from Referee 1 and will change the sentence accordingly to make it clearer.

**Line 80:** "is" instead of "does"?

**Response:** The appropriate change will be made

**Line 94:** Clean up unclear phrase "site-specific scale approach"

**Response**: This point was also reported by Referee 1 and we agree with the comments. We will revise the sentence and replace the 'site-specific scale approach', which is not appropriate for our multi-site approach.

**Line 97:** who is "they"?

**Response:** This overall last paragraph will be rephrased to increase clarity and to better reflect the aim of this study.

**Line 106:** "height" rather than "high".

**Response:** This will be corrected.

**Line 109:** 22% of what?

**Response:** of erosion "rate". This will be clarified.

**Line 120:** Who is "it"?

**Response:** Here we are talking about the retreat rate, we could make it more straightforward.

**Line 126:** Meaning of "porewater into"?

**Response:** The sentence will be corrected to: "Meltwater and groundwater (here defined as porewater in sandy coastal sediment).

**Line 134:** Delete "thawing.

**Response:** This will be removed

**Line 144:** GF/F filters have no nominal pore size.

**Response:** Thank you for pointing that now, it will be changed to : "DOC were collected using acid-cleaned 60 mL polypropylene syringes and filtered through pre-combusted 0.7 μm glass fiber (Whatman™ GF/F)"

**Line 169:** what is "the method"?

**Response:** the method used is the staRdom toolbox of Pucher et al., 2019 it is mentioned in the next paragraph. This can be reformulated to increase clarity.

**Lines 172-174:** Given this statistically questionable approach, it seems fair to state at least from how many computed indices you selected those three?

**Response:** The calculated indices included CDOM350, BIX, HIX, FI, SUVA, and SR. In this study, only CDOM350, HIX, and SUVA were reported, as they exhibited the most distinct variations among the different water sample types. The other indices were not included to streamline data presentation but can be provided. The sentence could be rephrased: "Different indices were explored, including CDOM350, BIX, HIX, FI, SUVA, and SR. However, only three were reported here to characterize the DOM pool, as they showed distinct variations among the different categories of samples (e.g., groundwater, melting water, massive ice, and seawater)."

**Line 185:** State sample size for PARAFAC.

**Response:** The PARAFAC model was built using over 250 samples collected from coastal Arctic and subarctic regions. This information will be added to the manuscript.

**Line 200:** Given the importance of "oxidative precipitation of amorphous Fe-hydroxides" in this manuscript, please provide more background information in the introduction.

**Response:** Agreed. Please see the comment above.

**Line 209:** Sp is a weird abbreviation for salinity.

**Response:** We respectfully disagree. "Practical salinity" is the appropriate term when salinity is derived from conductivity measurements using a probe calibrated with standard solutions, rather than from direct measurements with a salinometer. This terminology follows standard oceanographic conventions.

**Line 212:** Explain "tidal pumping effect" with more detail.

**Response:** Agree. We will add information on STE dynamics in the introduction, as proposed by Referee 1. A sentence such as: "*The tidal pumping effect, driven by the rising and falling tides, promotes the infiltration and recirculation of seawater through permeable beach sediments, influencing the salinity and oxygen dynamics of the beach groundwater (Santos et al., 2009)."*

**Line 216:** Over-saturation is not logically explained here. Contact with the atmosphere will drive a water body towards saturation, from over- or undersaturation.

**Response:** The oversaturation is likely explained by the in-situ production of oxygen by photosynthetic organisms during daylight.

**Lines 217-218:** You actually invoke mixing effects here, why not consider them as explanation for concentration gradients?

**Response:** See our comment above.

**Line 230:** Where would river water be on the LMWL?

**Response:** Based on the litterature on the MacKenzie, the isotopic signatures of surface water range from -24 to -10 and -115 to -18O, for $d^{18}O$ and $d^2H$, respectively. Meteoric waters that have undergone evaporation, including lakes and wetlands surface water display systematic enrichment in 18O and 2H and are mostly below the LMWL.

**Line 236:** Reconsider argument behind "suitable" here. You present no evidence for microbial transformations nor for mineral-organic interaction. Why is mixing of meltwater with river/sea water not an equally good explanation for concentration gradients?

**Response:** Please refer to our responses above. As previously mentioned, we will clarify the points raised and strengthen our arguments using the additional data available, even though we acknowledge their limitations. We would like to emphasize that our results are highly original and provide unique insights, despite the inherent constraints of working in such a challenging environment.

**Line 244:** The phrase "whatever the salinity" is used more often yet has unclear meaning.

**Response:** This will be rectified

**Line 254:** Subject behind "they" unclear.

**Response:** This sentence could be revised as such: "The median SUVA254 value of 2.4 mg C L-1 in the meltwater samples increased slightly to 2.9 mg C L-1 in the beach groundwater samples and further reached a median value of 3.4 mg C L-1 in seawater samples (Fig. 4D)."

**Lines 255-275:** This section is hard to understand. I feel DOC and CDOM/FDOM are treated like separate fractions of DOM (this comment applies almost throughout the R&D), but that is not a permissible strategy. DOC is an analytical proxy for DOM quantity, optical properties may allow conclusions about DOM-"quality" or "composition" or act as proxy for specific optically active moieties. I guess the main point of this text section is that decoupling of these parameters along an environmental gradient suggests changes in composition and thus points to active processes? If so, it remains unclear whether you actually invoke this to happen in your study. Present evidence more to the point.

**Response:**  Please refer to comments above. We agree that DOC, CDOM/FDOM should not be interpreted as distinct fractions of the DOM pool. Our intention was not to treat them as separate components, but rather to use DOC as a proxy for DOM quantity, and CDOM/FDOM to infer compositional or "quality" aspects of DOM. We will revise the text to avoid this ambiguity and to make it clearer that observed decoupling between DOC and CDOM reflects changes in DOM composition along the flow path. This is consistent with previous work (e.g., Del Vecchio and Blough, 2004) showing that DOC and CDOM can become uncoupled due to selective removal or transformation of optically active moieties, specifically in estuaries.

**Line 282:** Here and in the following text, please respect that you have not measured molecular weight per se but at best an optical proxy for molecular weight.

**Response:** We acknowledge that we have not directly measured molecular weight. As stated in Section 2.4 of the Methods, we use SUVA as an optical proxy to infer molecular weight. If necessary, we can clarify it again in this section.

**Line 284:** The phrase "less significant" is not meaningful.

**Response:** this could be replaced by "relatively small" (talking about the impact on the aromaticity).

**Line 286:** Photochemical process in darkness?

**Response:** Thank you for pointing that out, this will be rectified.

**Lines 287-289:** Seems speculative.

**Response:** Agree. Referee 1 also pointed out this point. We will reword the sentence as follows : "Additionally, the preferential precipitation of non-humic material and degradation of particulate organic matter could further enhance aromaticity by enriching the DOM pool with smaller, more aromatic compounds. " TOC will be reported in the Table 1.

**Lines 292-297:** Unclear evidence for two sources of C2.

**Response:** To clarify, we will revise the text to better explain the rationale for interpreting the stability of C2 as potentially reflecting contributions from both permafrost-derived and autochthonous sources. While we cannot directly trace the origin of C2 in this study, previous research has shown that protein-like fluorescent DOM (C2) can originate from both terrestrial (e.g., permafrost or riverine export) and microbial/algal activity. Given the simultaneous evidence of DOM degradation along the flow path and the stability of C2, we interpret this as consistent with a dynamic balance between loss and in situ production. We will add references and clarify this interpretation in the revised manuscript.

**Line 299:** Seems weird to argue for "affinity" differing among analytical measures.

**Response:** We intended to highlight that the observed behavior of DOC and CDOM differed in response to Fe-hydroxide precipitation, suggesting that the components detected by each proxy may interact differently with Fe-oxides. We will revise the sentence to clarify this point and to avoid implying that the analytical measures themselves have chemical affinity.

**Lines 302-304:** I remain a bit surprised that iron data is not presented in more detail. Sentence 303-304 not understandable.

**Response:** We agree with this comment, total dissolved iron and reactive sediment Fe will be added in the revised manuscript.

**Lines 311-316**: Seems the Fe-effect experiences some sort of saturation at least in the experiment? Would this not be relevant in the beach itself as well?

**Response:** The aim of the experiment was indeed to reach saturation, as our objective was to highlight the differences in affinity between DOM and DOC for reactive Fe-hydroxides under controlled conditions. Rather than replicating in situ beach conditions (based on total dissolved Fe concentrations), we deliberately exaggerated the process to simulate Fe-coated sediment scenarios and better understand the potential for Fe-DOM interactions. While these saturation levels may probably not reflect natural beach concentrations, they offer valuable insight into the mechanisms that could occur under iron-coated rich conditions within the STE.

**Fig. 3:** So, what do we really learn from these data? I cannot see that from the text.

**Response:** Please refer to our different responses above.

**Fig. 4:** Consider ordering sites along the flowpath from meltwater to seawater. Font size generally much too small (also applies to other figures).

**Response:** We agree with this comment and will make the appropriate changes.